Disentangling direct and indirect effects of local temperature on abundance of mountain birds and implications for understanding global change impacts

Ceresa Francesco 1 francesco.ceresa01@gmail.com
Kranebitter Petra 1
http://orcid.org/0000-0002-0952-2089 S. Monrós Juan 2
Rizzolli Franco 1
http://orcid.org/0000-0002-7643-4652 Brambilla Mattia 3 4 5
1 Museum of Nature South Tyrol , Bolzano, BZ , Italia
2 Universidad de Valencia , Valencia , Spain
3 Museo delle Scienze , Trento , Italia
4 Fondazione Lombardia per l’Ambiente , Milano , Italia
5 Dipartimento di Scienze e Politiche Ambientali, Università degli Studi di Milano , Milano , Italia
Pimm Stuart
Electronic publication date: 2021 Dec 3
Publication date: 2021
Volume: 9
Electronic Location ID: e12560
Received 2021 Jul 29; Accepted 2021 Nov 7
Copyright: © 2021 Ceresa et al.
Copyright year: 2021
Copyright holder: Ceresa et al.
License: This is an open access article distributed under the terms of the Creative Commons Attribution License, which permits unrestricted use, distribution, reproduction and adaptation in any medium and for any purpose provided that it is properly attributed. For attribution, the original author(s), title, publication source (PeerJ) and either DOI or URL of the article must be cited.
License URL: https://creativecommons.org/licenses/by/4.0/

Keywords: Elevational gradient, Bird ecology, Thermal niche, European Alps, High-elevation species

Funding: Research fund of the Museums of South Tyrol CUP H53C17000260005 Department of Innovation, Research and University of the Autonomous Province of Bozen/Bolzano The present study has been financed by the Research fund of the Museums of South Tyrol, within the project ‘The distribution and conservation status of birds in South Tyrol’, CUP H53C17000260005. The Department of Innovation, Research and University of the Autonomous Province of Bozen/Bolzano covered the Open Access publication costs. The funders had no role in study design, data collection and analysis, decision to publish, or preparation of the manuscript.

==============================
Unravelling the environmental factors driving species distribution and abundance is crucial in ecology and conservation. Both climatic and land cover factors are often used to describe species distribution/abundance, but their interrelations have been scarcely investigated. Climatic factors may indeed affect species both directly and indirectly, e.g., by influencing vegetation structure and composition. We aimed to disentangle the direct and indirect effects (via vegetation) of local temperature on bird abundance across a wide elevational gradient in the European Alps, ranging from montane forests to high-elevation open areas. In 2018, we surveyed birds by using point counts and collected fine-scale land cover and temperature data from 109 sampling points. We used structural equation modelling to estimate direct and indirect effects of local climate on bird abundance. We obtained a sufficient sample for 15 species, characterized by a broad variety of ecological requirements. For all species we found a significant indirect effect of local temperatures via vegetation on bird abundance. Direct effects of temperature were less common and were observed in seven woodland/shrubland species, including only mountain generalists; in these cases, local temperatures showed a positive effect, suggesting that on average our study area is likely colder than the thermal optimum of those species. The generalized occurrence of indirect temperature effects within our species set demonstrates the importance of considering both climate and land cover changes to obtain more reliable predictions of future species distribution/abundance. In fact, many species may be largely tracking suitable habitat rather than thermal niches, especially among homeotherm organisms like birds.

Introduction

Climate and land-use/land-cover strongly affect the distribution and abundance of species and many ecosystems functions (e.g., Sala et al., 2000). To correctly understand the impact of global change, and in particular of climate and land-use modification, unravelling the environmental factors determining species distribution and abundance is crucial (Rodríguez et al., 2007; Péron & Altwegg, 2015; Engler et al., 2017). While this is a key issue across ecosystems and realms (Mantyka-Pringle, Martin & Rhodes, 2012), for some groups, such as birds (e.g., Jongsomjit et al., 2013), and some environments, such as mountains (Dirnböck, Essl & Rabitsch, 2011), disentangling those effects is of utmost importance to address urgent conservation challenges (Chamberlain et al., 2013). Unfortunately, studies explicitly addressing both land-use and climate change effects are relatively scarce (e.g., Pearce-Higgins et al., 2015), and there are few studies that have really tried to quantitatively partition the interactive effects of climate and land-use on target species.

Mountains are characterized by a strong variation across short distances of abiotic factors such as temperature, precipitation, atmospheric pressure, slope and exposure. This leads to a high variety of habitats and species occurring within relatively small areas and stratified across elevational gradients (Körner & Ohsawa, 2006; Cadena et al., 2012). This often results also in patchily distributions of species that are strictly connected to very specific elevation ranges, and also in high endemism levels (Ruggiero & Hawkins, 2008; Cadena et al., 2012). Mountain ecosystems are consequently extremely valuable in terms of biodiversity and conservation (Myers et al., 2000; Körner & Ohsawa, 2006; Boyle & Martin, 2015), but species specialist of those ecosystems are highly threatened by climate change (e.g., Goodenough & Hart, 2013; Brambilla et al., 2017; De Gabriel Hernando et al., 2021), because of their adaptation to local conditions (Cheviron & Brumfield, 2012). Among birds, species distribution/abundance along elevational gradients have been investigated in several studies (e.g., Chamberlain et al., 2016; Frey, Hadley & Betts, 2016; Elsen et al., 2017; Jähnig et al., 2020), considering factors such as local climate, vegetation characteristics and topography as explanatory variables. Studies on factors driving occurrence or abundance along environmental gradients provide highly valuable information, but the interplay between climate and vegetation in determining mountain bird distribution/abundance is still unclear. In fact, these factors are usually considered independently in data analyses, without estimating the indirect effects deriving from the causal relationships between climate and vegetation (but see Duclos, DeLuca & King (2019) for a set of forest bird species). As a consequence, the degree to which climate impacts on species’ occurrence or abundance directly (by affecting, e.g., thermoregulation, nest site selection, breeding phenology; Martin, 2001; Leech & Crick, 2007; Barnagaud et al., 2011; Bison et al., 2020) or indirectly, through e.g., its influence on vegetation structure and composition, is largely unknown. This is a highly relevant issue for ecology and conservation, especially because it is key to predict climate change effects on bird populations: species distribution changes and elevational shifts indeed are often interpreted as a thermal niche tracking but, if climate effects are mostly or exclusively indirect, such distribution changes will be rather a consequence of habitat tracking. In this case, predictive distribution modelling should account for future changes in vegetation structure and composition due to climate change and other (possibly interacting) factors such as human land use. Knowledge about the relative magnitude of direct and indirect climate effects would also allow to propose more effective conservation strategies and management measures, or to improve the existing ones.

In this study, we aimed to assess and disentangle the direct and indirect effects of local temperature on bird abundance. We focused on breeding birds in the European Alps, along a wide elevational gradient (1,300–2,700 m asl) including montane and subalpine forests, the treeline habitat and alpine grasslands and rocky areas at the highest elevation. To our best knowledge, there are no previous studies aimed at quantitatively disentangling direct and indirect effects of local climate on species distribution/abundance encompassing such a wide elevational gradient. We expected a clear indirect effect of local temperature in all bird species: across such a wide elevational gradient, with forest gradually transitioning into alpine grassland, a strong effect of local climate on vegetation should be supposed, as climate is the main determinant of vegetation characteristics, and a definitive effect of vegetation on birds can be expected due to the striking differences in vegetation structure and composition along the gradient. Concerning the direct effect of local temperature, we expected strong interspecific differences, rather than generalized patterns, as a consequence of species-specific thermal niches and different degree of overlap with the climatic conditions of the study area. More specifically, we expected mountain generalist species to be more associated with warmer sites, given the harsh climate of our inner-Alpine study area (Adler et al., 2015). Conversely, for mountain specialists (e.g., water pipit Anthus spinoletta), we expected a preference for relatively cold microclimates (Chamberlain et al., 2013; Brambilla et al., 2016, 2017; mountain specialist vs generalists were distinguished following Lehikoinen et al., 2019).

Materials and Methods

Data collection

Data were collected as previously described in Ceresa et al. (2020a). Specifically, fieldwork took place in the central-eastern Italian Alps (Wipptal, South Tyrol), immediately south of the main Alpine watershed, within a 3,400 ha-wide area (46.96° N, 11.50° E). This area spans from approximately 1,300 m asl (just above the bottom valley) to more than 2,700 m asl at the highest peaks. At lower elevation, mountainsides are mostly covered by woodlands, dominated by spruce Picea abies and European larch Larix decidua. Above the timberline (approximately 2,000 m asl, but highly variable), wide areas are covered by bushes (mainly Rhododendron spp.) and scattered larches, whereas alpine grasslands, rocks and scree slopes characterize the upper elevation belt.

Birds were surveyed during the breeding season of most local breeding species (May–July) in 2018, by means of point counts (N = 109, Fig. 1) carried out by expert ornithologists. The rugged topography, and the consequent low accessibility of some areas, made a random selection of the sampling points unfeasible. For this reason, we carried out the point counts along accessible transects, often along footpaths; we chose the location of each point based on a minimum distance of 200 m to the previous one, in order to avoid potential replicated counts of the same individuals (Chamberlain et al., 2016).

Figure 1 Study area.

Distribution of the sampling points in the study area (central-eastern European Alps, Italy).

At each sampling point, the surveyor recorded all birds observed or heard during 10 min within a 100 m radius. All points were visited three times (survey sessions: 30 May–6 June, 19–23 June and 13–18 July). The survey timing allowed excluding the main migration periods of locally breeding species. Counts were carried out in the morning, between 5:30 and 11:30, and avoiding poor weather conditions (i.e., rain or strong wind).

Vegetation and other land cover characteristics within a 100 m radius were recorded at each sampling point by the surveyor, with the aid of detailed aerial photos (as described in Ceresa et al. (2020a)). These variables included the percentage cover of tree canopy (i.e., vegetation higher than 2 m), bushes (woody vegetation lower than 2 m, mostly represented by Rhododendron shrublands), grassland (areas with no canopy and covered by grassland vegetation) and rocks/scree (emerging bedrock or scree-covered patches). The percentage cover of each variable was thus visually estimated in the field to the nearest 5%. Variables covering less than 5% of the plot surface (as defined by the 100 m radius) were assigned a 1% cover value. We also recorded information about woodland composition, by visually estimating the proportion of tree canopy cover occupied by each arboreal species (excluding rare species, i.e., representing less than 20% of the tree canopy cover).

In order to obtain climatic information at the site level, at each sampling point we placed an iButton data logger (models DS1921G and DS1922L; Maxim Integrated, San Jose, CA, USA), which recorded temperatures hourly during the entire study period. Loggers were placed at the ground level and were protected from the direct solar radiation by means of a white plastic panel. Further details about loggers’ placement and use, as well as the temperature trends recorded at each point during the study period, are provided in Ceresa et al. (2020a). As we collected such fine-scale temperature values, we did not consider topographic variables in our study. In fact, the effects of topography on bird abundance are very likely to be largely indirect, through a strong influence on microclimate (especially temperature) and, consequently, also on vegetation. Our fine-scale temperatures were indeed clearly related to both elevation and aspect, as expected (see Fig. S1). Relevant direct effects of topography on bird abundance (non-mediated by local temperature and vegetation) are indeed hardly to occur, especially for elevation and aspect.

Statistical analysis

We used structural equation modelling to disentangle the direct effect of microclimate on bird abundance and the indirect effects of local temperature, mediated by vegetation characteristics. Structural equation modelling consists in a multivariate linear regression analysis, which allows assessing the effect of a predictor on a dependent variable through mediating variables. Therefore, following this approach, it is possible to model simultaneously both direct and indirect effect of a predictor on a dependent variable (Rosseel, 2012). In the model structure we adopted, variables describing vegetation characteristics represented the mediating variables in the indirect pathways relating local temperature and bird abundance (see the path diagrams in Fig. 2). Models were fitted using the sem function of package lavaan version 0.6-7 (Rosseel, 2012) in program R version 3.5.3 (R Core Team, 2019); we used the Satorra-Bentler maximum likelihood test statistics, which provides model fit measures that are robust to data non-normality (Satorra & Bentler, 1994). We considered standardized path coefficient as significant at p ≤ 0.1, to reduce the risk of type II error associated with the adopted statistical approach (Shipley, 2016). An indirect effect was considered as significant when all single path coefficients of the indirect pathway were significant at p ≤ 0.1. We compared direct and indirect effects of local temperature by comparing the standardized coefficients of direct and indirect pathways.

Figure 2 Relationship among temperature, vegetation characteristics and bird abundance.

Relationship among temperature, vegetation characteristics (PC1 and PC2) and bird abundance according to structural equation modelling in four of 15 investigated bird species breeding in a mountain area. The reported values are standardized regression coefficients and the asterisk indicate a significant effect (p ≤ 0.1).

To describe vegetation characteristics in our models, we used two principal components describing overall 60% of the variability in the vegetation data collected in the field (PC1 = 40%, PC2 = 20%). Principal component analysis was carried out using the prcomp function in the base R package. The first component was positively associated with forested areas and negatively with open areas (grassland and rocks), while the second component described a gradient in woodland composition (it was negatively associated with larch, one of the two dominant arboreal species), as well as the degree of grassland cover in open areas (positive association with rocks, negative with grassland; see Table 1). As local climatic predictor, we used the mean temperature calculated for each sampling point across the entire study period, because it allows describing the general climatic conditions at a location (Virkkala et al., 2008; Stralberg et al., 2009). Mean temperatures during the study period varied across the study area between 6.6 °C and 14.8 °C, and obviously showed a clear trend along the elevation gradient (Fig. S1). As a measure of bird abundance, for each species we used the maximum number of individuals detected at each sampling point during the three sampling sessions. Unlike other studies dealing with mountain species’ ecology, but not aimed at quantitative disentangling direct and indirect climate effects (e.g., Chamberlain et al., 2013; Ceresa et al., 2020a), we considered the points recorded over the complete elevational gradient for all species. This implied that, for each species, we included also areas with clearly unsuitable vegetation characteristics. Given that vegetation characteristics are largely determined by local climate, the selection of sub-samples of the sampling points based on vegetation characteristics (as frequently done in other studies with different aims), would have biased the estimation of the indirect temperature effects on bird abundance (likely leading to strong underestimation of such effects). We only considered those species detected at least once at a minimum of 20 sampling points, i.e., 15 species. These species (listed in Table 2) show a variety of different habitat preferences and specialization, as well as different nesting habits and migration strategies, therefore our results are based on a highly representative species set.

Table 1 Relationship between vegetation variables collected in the field and PCA components used to descibe vegetation.

Relationship between the two principal components (used to describe vegetation structure and composition in structural equation models) and each land cover category.

Land cover category	PC1	PC2	
Bushes (% cover)	0.053	−0.663	
Rocks/scree (% cover)	−0.328	0.424	
Tree canopy (% cover)	0.572	0.340	
Undergrowth (% cover)	0.318	−0.163	
Grassland (% cover)	−0.493	−0.368	
Larches (% of total canopy cover)	0.468	−0.321	

Table 2 Direct and indirect effects of local climate on bird abundance.

Effects of vegetation (PC1 and PC2) and temperature on breeding bird abundance according to structural equation modelling (p-values in parentheses), and variance in bird abundance explained (R2) for each species. Effects are reported as standardized coefficients. Only significant effects are reported (p ≤ 0.1). An indirect effect is considered significant when all path coefficients of the indirect effect path are significant at p ≤ 0.1. Model structure is shown in Fig. 2.

Species	PC1	PC2	Temperature (direct effect)	Indirect temperature effect through PC1	Indirect temperature effect through PC2	R2	
Anthus spinoletta	−0.716 (0.000)	−0.145 (0.040)		−0.387 (0.000)	0.023 (0.191)	0.44	
Certhia familiaris	0.485 (0.000)	0.349 (0.000)		0.262 (0.000)	−0.055 (0.080)	0.37	
Erithacus rubecula	0.460 (0.000)	0.341 (0.001)	0.244 (0.002)	0.249 (0.000)	−0.054 (0.098)	0.46	
Fringilla coelebs	0.766 (0.000)			0.414 (0.000)		0.63	
Lophophanes cristatus	0.499 (0.000)	0.283 (0.005)	0.203 (0.025)	0.270 (0.000)	−0.045 (0.177)	0.44	
Oenanthe oenanthe	−0.298 (0.001)			−0.161 (0.003)		0.19	
Periparus ater	0.704 (0.000)	0.291 (0.000)	0.162 (0.008)	0.381 (0.000)	−0.046 (0.086)	0.68	
Phylloscopus collybita	0.328 (0.005)			0.178 (0.006)		0.17	
Poecile montanus	0.451 (0.000)	−0.265 (0.004)		0.244 (0.000)	0.042 (0.095)	0.29	
Prunella modularis	0.482 (0.000)	−0.164 (0.051)		0.261 (0.000)	0.026 (0.167)	0.34	
Pyrrhula pyrrhula	0.440 (0.000)	0.267 (0.000)	0.132 (0.069)	0.238 (0.000)	−0.042 (0.088)	0.31	
Regulus regulus	0.301 (0.000)	0.449 (0.000)	0.289 (0.002)	0.163 (0.000)	−0.029 (0.071)	0.41	
Sylvia atricapilla	0.309 (0.005)		0.187 (0.096)	0.167 (0.004)		0.21	
Turdus torquatus		−0.332 (0.007)			0.053 (0.123)	0.13	
Turdus viscivorus	0.199 (0.058)		0.186 (0.029)	0.107 (0.060)		0.11	

We evaluated model goodness-of-fit by a chi-square fit statistic, which indicates a significant lack of fit when p < 0.05 (Hooper, Coughlan & Mullen, 2008). We also used other fit metrics commonly adopted in structural equation modelling: Comparative Fit Index (CFI), Tucker-Lewis Index (TLI), root mean square error of approximation (RMSEA) and standardized root mean square residual (SRMR). A good model fit is indicated by CFI and TLI > 0.95, RMSEA < 0.07 and SRMR < 0.05 (Hooper, Coughlan & Mullen, 2008). Using and reporting such a variety of fit metrics is recommended in structural equation modelling, because each one reflects a different aspect of model fit (Crowley & Fan, 1997; Hooper, Coughlan & Mullen, 2008). In order to test for spatial autocorrelation, we calculated Moran’s I for model residuals using ArcGis 10.8 (ESRI, Redlands, CA, USA). Moran’s I values > 0.3 are considered as relatively large (Lichstein et al., 2002).

Results

For all species, structural equation models indicated a significant effect of vegetation characteristics on bird abundance (Table 2). Out of 15 species, PC1 had a significant effect in 14 and PC2 in 10 species. Given the significant effect of temperature on both PC1 (β = 0.54, p < 0.001) and PC2 (β = −0.16, p = 0.063), indirect pathways were always significant in case of significant effect of a principal component on bird abundance. Indirect temperature effects were indeed found in all species, while direct temperature effects occurred in seven species (Table 2). When present, direct temperature effects were always positive and showed a highly variable relative importance compared to indirect effects (Table 2). Variance in bird abundance explained by structural equation models ranged between 11% and 68%, with only four species below 20% (Table 2). For all species, models fitted the data well according to the chi-square test (χ2 = 1.265, df = 1, p = 0.261) and all the other fit metrics (CFI and TLI > 0.95, RMSEA < 0.07 and SRMR < 0.05). For seven species, Moran’s I test indicated the lack of spatial autocorrelation (Moran’s I range: −0.037 to 0.065; p > 0.05; see Table S1). In the other species the test was significant, but with very low or low Moran’s I values (approx. 0.1–0.2; Table S1), indicating a weak clustering that is very unlikely to affect results. The only exception is represented by the common chiffchaff Phylloscopus collybita, which showed a relatively high spatial autocorrelation (Moran’s I = 0.414); the results for this species should be cautiously considered.

Discussion

Disentangling direct and indirect effects of climate on species occurrence and abundance is crucial to assess distribution and population drivers, and to evaluate the potential impacts of climate change on wild species (de Chazal & Rounsevell, 2009). The correlative link between climate and species (especially for homoeotherm animals) often reflects an indirect effect, which is actually mediated by other factors associated with climate, such as habitat characteristics or the availability of key resources (Brambilla et al., 2019a). Here, we investigated the drivers of bird abundance in a mountain context, where climate and habitat effects are intermingled, explicitly distinguishing between direct and vegetation-mediated impacts of local temperatures.

As expected, we observed a significant indirect effect of local temperature via vegetation characteristics on the abundance of all the species we considered. This suggests that such a pattern could be generalized, or at least very common in bird communities of temperate mountain regions, given the wide ecological spectrum represented in our species set. Temperatures at sampling sites had a significant effect on both principal components describing vegetation, with a stronger influence on the most important one (PC1), which accounts for the difference between forested and open areas (Fig. 2). This is not surprising, as climatic factors are crucial in determining the treeline position, jointly with several additional factors such as human land use and geomorphology (Körner & Paulsen, 2004; Chauchard et al., 2010; Leonelli et al., 2011). The observed effects of the two vegetation principal components on bird abundance are fully consistent with the known habitat preferences of each species (e.g., Brichetti & Fracasso, 2007, 2008; Chamberlain et al., 2016). Unlike indirect effects, direct temperature effects only occurred in some species, consistently with our expectation of strong interspecific differences and lack of generalized patterns.

Direct effects of local climate for birds in the Alps

All the seven species showing significant direct temperature effects (see Table 2) are woodland or shrubland birds and are not mountain specialists. The distribution of some of these species in Italy widely overlaps with the main mountain chains, but they cannot be considered as mountain specialists because they breed also in hilly and low elevation areas, and in nearby countries they are also widely distributed across lowlands (coal tit Periparus ater, crested tit Lophophanes cristatus, Eurasian bullfinch Pyrrhula pyrrhula, goldcrest Regulus regulus, mistle thrush Turdus viscivorus; species distributions are available in BirdLife International (2021)). Given the positive direct temperature effect on their abundance, at least part of the woodland within our study area is likely colder than the thermal optimum of these species; our inner-Alpine study area is indeed characterized by an especially severe climate, with long-lasting snow cover (Adler et al., 2015). Such temperature effect could be emphasized by the early breeding of these species: warmer sites likely provide better conditions early in the reproduction season, especially for what concerns the availability of key poikilothermic prey (invertebrates). Two other species showing a positive direct temperature effect (Eurasian blackcap Sylvia atricapilla and European robin Erithacus rubecula) are generalist species with very broad habitat niche, and are very common at low and middle elevations; our study area is probably located towards the coldest extreme of their thermal niche.

In the two open habitat species for which we obtain a sufficient sample for the analysis (water pipit and northern wheatear Oenanthe oenanthe), we found a significant negative temperature effect on abundance, but exclusively indirectly via vegetation. This suggests that, at least in inner Alpine areas like our study site, maintaining and correctly managing high-elevation pastures and grasslands would probably represent effective conservation measures for these species, as proposed also in other studies (e.g., Chamberlain et al., 2013; Brambilla et al., 2018, 2020). Differently from what we expected, our results indeed did not show a direct temperature effect for the water pipit, a mountain-specialist grassland bird that had been reported to be associated with cold climates in other studies focusing on species occurrence (e.g., Brambilla et al., 2017, 2019a), and to experience variation in breeding success according to nest-site orientation (Rauter, Reyer & Bollmann, 2002). The most likely explanation for such differences is that, in this cold, inner-Alpine area, the climatic conditions of alpine grasslands are close to the thermal optimum of this species. On the other side, the previous studies reporting an effect of local climate on this species’ occurrence were carried out at a much larger scale, including also low-elevation, warmer massifs (Brambilla et al., 2017), or took place in more southern and definitely warmer mountain chains (i.e., Apennines; Brambilla et al., 2019a, Brambilla et al., 2020). The large differences in temperature ranges between our and those previously investigated study areas probably explains the differences with our results, which revealed no direct effect of temperature on abundance (besides the different focus on occurrence vs. local abundance, and the different statistical approach). This highlights the importance of spatial scale when studying the influence of climatic factors on species distribution and abundance, as different or even contrasting results may arise from analyses carried out at different scales (e.g., Franklin et al., 2013; Brambilla et al., 2019a).

While the proportion of variance in bird abundance explained by our models was good for most species (see Table 2), the proportion of unexplained variance suggests the more or less important effect of additional factors not considered in this study, such as spatial variation in predation intensity and other biotic interactions, microhabitat characteristics, and possibly also terrain slope in some species (see, e.g., Thompson, 2007; Sherry et al., 2015; Freeman & Montgomery, 2016; Brambilla et al., 2019a). We could not consider also the possible effect of precipitations on bird abundance, because we did not collect precipitation data in the field. However, over such a restricted study area the spatial variation of precipitation is likely to be affected almost only by the reduction of temperatures with elevation (through orographic effect), which should be strictly related to our fine-scale temperature data. Therefore, precipitation data would likely add only limited information to our models.

Implications for research and conservation

Our results are largely consistent with those reported by Duclos, DeLuca & King (2019), who investigated a set of forest bird species breeding in a North American mountain area and found an indirect effect of climate via vegetation in all species, while directs effects occurred only in part of the species and strongly varied in their magnitude. This suggests that such climate-species relationship patterns may be widespread in mountain bird communities, at least in temperate regions and in areas where the number of high-elevation specialists is often very low, as in our study system. This should be taken into account when investigating and predicting future bird distribution and abundance. This could be particularly relevant for generalist mountain birds, as in many cases they may be largely tracking suitable habitat, rather than thermal niches, as our results suggest for several species inhabiting different habitats and different elevation belts. Consequently, accounting also for future changes in vegetation structure and composition would likely allow more reliable predictions than using climatic factors only; this is a complex task, as both human land use and climatic factors should be considered to forecast vegetation changes. In addition, vegetation changes may lag largely behind climate change (Iverson, Prasad & Matthews, 2008; Stralberg et al., 2015): in e.g., the northeastern United States tree species shift has been forecasted to occur in some centuries (Wang et al., 2016). Such a complex and context-dependent scenario (species-specific direct and indirect climate effects, influence of human land use) may help to explain why elevational range shifts of mountain bird distribution assessed until now are poorly consistent across different studies in both magnitude and direction (i.e., uphill/downhill; see Scridel et al. (2018)), in spite of the global scale temperature increase.

While indirect climate change effects via vegetation may be strongly delayed or influenced by human activities, direct effects are likely to impact bird population rapidly, by affecting, e.g., breeding phenology, prey availability, survival and breeding success (e.g., Rodriguez & Bustamante, 2003; Dunn, 2004; Wesołowski et al., 2016; Bison et al., 2020; Ceresa et al., 2020b; Strinella et al., 2020). According to our results, at least in the innermost part of the European Alps the ongoing temperature increase may favour some generalist woodland and shrubland bird species (as suggested, e.g., by Solonen, 2005; Scridel et al., 2017). This could occur due to wider woodland areas with adequate local climatic conditions for these species, but also because of a possibly longer breeding season (Bison et al., 2020). According to Ceresa et al. (2020a), in our study area European robins select warmer areas for breeding during the first part of the reproductive season, thus higher spring temperatures would likely allow start breeding earlier across wider areas, with a consequently larger time window available for subsequent reproduction attempts.

The indirect effect of local climate we found in the two open habitat species here considered highlights the importance of conservation and management of Alpine grasslands for those species. Uphill treeline shift and shrubland expansion have been recorded in several mountain ranges, including the European Alps, and besides temperature increase they are often strongly promoted by land abandonment (Gehrig-Fasel, Guisan & Zimmermann, 2007; Myers-Smith et al., 2011). Therefore, habitat loss for these species needs to be counteracted by maintaining cattle grazing in mountain pastures, managed in an extensive way to avoid overgrazing, which leads to grassland and soil degradation (Garcia-Pausas et al., 2017) and can be detrimental to open-habitat bird species (Brambilla et al., 2020). In addition, the construction of new touristic infrastructures such as ski-pistes should be limited, because they negatively affect high-elevation grassland bird communities (Rolando et al., 2007; Caprio et al., 2011) and are predicted to increasingly overlap with the distribution of high-elevation specialist birds as a consequence of climate change (Brambilla et al., 2016). Other bird species connected to high-elevation open areas could be affected by climatic factors also directly, e.g., the white-winged snowfinch Montifringilla nivalis (Brambilla et al., 2019b), and unfortunately we did not obtain a sufficient sample for this species and other locally breeding high-elevation specialists (alpine accentor Prunella collaris, alpine chough Pyrrhocorax graculus). However, also in case of direct climate effects the aforementioned conservation measures would be recommendable, in order to try buffering against climate change. Given the high terrain complexity of mountain ranges, across large areas of suitable habitat some sites may maintain adequate microclimatic conditions for cold-associated species, possibly acting as local ‘refugia’ despite temperature warming at larger scale (Morelli et al., 2016). Therefore, avoiding habitat loss and degradation as previously described could increase the future amount of refugia maintaining both vegetation and microclimatic suitable characteristics. Furthermore, correctly managed cattle grazing could provide/maintain micro-habitats suitable for foraging for species potentially vulnerable to earlier snowmelt (Brambilla et al., 2018).

Conclusions

The impact of climate and land-use/land-cover changes are among the major threats to biodiversity worldwide, and will exacerbate in the future decades. Properly addressing the nature of their effects on wild species is key to understanding impacts and developing conservation strategies. In this study, considering a set of bird species along a broad elevational gradient, we found a generalised indirect effect via vegetation of local temperatures on birds abundance, while direct effects were less common and were found in some mountain generalist birds. Our work provides an example of disentangling causes and effects when dealing with the combined impact of habitat and local climate on target organisms; a similar framework may be used to address effects and impacts on many other ecosystems, promoting a deeper understanding of species’ response to habitat and climate and the relative changes.

Supplemental Information

Supplemental Information 1 Data used in structural equation models.

Bird abundance, vegetation and mean temperature at each sampling point.

Click here for additional data file.

Supplemental Information 2 Relationship between mean temperatures and elevation (left) in the study area, and the same relationship after accounting for aspect (right).

Elevation and aspect jointly explain 56% of variation in mean temperatures according to multiple linear regression (mean temperature ~ elevation + aspect), and both show a significant effect (elevation: β = −1.3, p < 0.0000; aspect: β = −0.3, p = 0.007).

Click here for additional data file.

Supplemental Information 3 Vegetation/land cover variables collected in the field at each sampling point.

Click here for additional data file.

Supplemental Information 4 Results of Moran’s I test for all considered species.

Spatial autocorrelation in model residuals for each species, according to Moran’s I test.

Click here for additional data file.

We thank Prof. Emilio Barba (Institute Cavanilles of Biodiversity and Evolutionary Biology-University of Valencia) for providing part of the temperature loggers. We are very grateful to D. Chamberlain and an anonymous reviewer for helpful comments on a first draft of the manuscript.

Additional Information and Declarations

Competing Interests

Author Contributions

Data Availability

The authors declare that they have no competing interests.

Francesco Ceresa conceived and designed the experiments, performed the experiments, analyzed the data, prepared figures and/or tables, authored or reviewed drafts of the paper, and approved the final draft.

Petra Kranebitter conceived and designed the experiments, authored or reviewed drafts of the paper, and approved the final draft.

Juan S. Monrós conceived and designed the experiments, authored or reviewed drafts of the paper, and approved the final draft.

Franco Rizzolli conceived and designed the experiments, performed the experiments, authored or reviewed drafts of the paper, and approved the final draft.

Mattia Brambilla conceived and designed the experiments, analyzed the data, prepared figures and/or tables, authored or reviewed drafts of the paper, and approved the final draft.

The following information was supplied regarding data availability:

The complete data used in structural equation models, i.e., bird abundancies, mean temperatures and vegetation (as PCA components) for each sampling point is available in the Supplemental File.

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
