# Peer review of "Disentangling direct and indirect effects of local temperature on abundance of mountain birds and implications for understanding global change impacts"

_PeerJ, doi:10.7717/peerj.12560_

## Round 0.1 · original submission · Major Revisions

As you can see, the reviewers have raised substantial issues that we need you to address. I look forward to your revisions.

·

Basic reporting

The paper is generally clear and the English is good (there are a few grammatical errors that could be corrected at the editing stage). The structure is correct, figures and data are in order.

I have a few minor comments on the context and background.

1. L37-39 – This is a bit confusing – studies addressing both land-use and climate change effects over time (i.e. with long-term data) are scarce, but there are many studies (like this one) that consider both of these under a space-for-time substitution approach (e.g. those cited on L50-51). I think you need to expand a bit here – “there are few studies that have really tried to partition the interactive effects of climate and land-use” would be more accurate.
2. L48 – Give a reference to support the statement that specialist species of these ecosystems are highly threatened. The implication is that Cheviron and Brumfield says this, but in fact, this citation is relevant to only the second part of the sentence.
3. L72-74 – as per comment 1, I do not agree with this statement. There have been several studies that have considered the relative contributions of direct climate effects (i.e. distributions modelled in relation to temperature and precipitation) and indirect effects (as expressed by vegetation zones along the elevation gradient), many of which are cited. The goal here is a little more subtle than described – previous studies have looked at these effects, but I guess you would argue that your approach better addresses the interaction between them and hence can better measure their independent effects – so say this!
4. L122-127 – the approach of using temperature loggers in conjunction with point counts is pretty much the same as used by Jähnig et al. 2020 (DOI 10.1007/s10336-020-01778-5). This paper also has some highly relevant results. I am surprised it is not cited.
5. In general, I don’t think that the paper is placed in the correct context of the existing literature. You need to be clear on what the paper replicates and what it provides that is new.

Experimental design

I have some major comments on this section.

1. By using all points for each species, you will be including habitats that are completely unsuitable. For example, we know that Crested and Coal Tits require trees for nesting and they are not found in open treeless areas. Similarly, a Water Pipit will not occur in the middle of a forest. Absences in unsuitable habitat in these cases may be seen as ‘naughty naughts’ or ‘redundant zeros’ (see Zuur et al. 2009 Mixed Effects Models and Extensions in Ecology with R). I suspect the models are to some extent just describing the well-known division between ‘open’ and ‘closed’ habitats rather than telling us anything more interesting. Several papers that you cite have divided the alpine bird community into open and closed habitat species before analysis in order to avoid this issue. I think what you have done is worth keeping, but I think follow-up analyses, restricting each species to it’s likely potential nesting habitat, is needed.
2. Several published papers considering bird-habitat associations in alpine landscapes have included topographic variables (slope and aspect), and in some cases, these have proven to be important predictors. Moreover, it seems plausible that they may have interaction effects with temperature and thus are likely to be highly relevant to the current study. There is no reference at all to the potential importance of topography. These data can be readily extracted at a pretty high resolution from existing DTMs. I think they need to be included.
3. Spatial autocorrelation may be an issue and should be tested for (again, most of the cited papers analysing alpine species distributions have tested for spatial autocorrelation, although it seems rare that there are important effects).

Validity of the findings

Some minor comments on the Discussion.

1. L207-208 – What is meant by ‘high interspecific variability’. There was variation in significance, but for those species that showed a significant effect, it was always positive. Only significant effects are reported in Table 2, so we can’t tell whether effects were generally positive (whether significant or not). Re-word this sentence.
2. L212-213 – I would argue that there is only one mountain specialist, the water pipit (possibly also the wheatear, although this can occur in a wider range of open habitats). It needs to be stressed that you cannot really conclude anything about mountain specialists more broadly.
3. L230-233 – I agree with this statement, but you should acknowledge that this is hardly original. It has been proposed several times (by many of the papers cited here). Rather you should say that the work supports previous research that has suggested grazing as a conservation tool to maintain open habitats in the face of shifting treelines.
4. L235-247 – There are better papers to cite that deal specifically with the Water Pipit. In particular, Rauter et al. 2002 (Ibis 144, 433–444) considers how micro-climate and topography interact to affect nesting success (this is also relevant to EXPERIMENTAL DESIGN comment 2). Related papers from the same group should also be considered.
5. L251 – including topography.
6. L284-286 – Again, this statement supports several other studies and I think this should be acknowledged.

Additional comments

No comment

Reviewer 3 ·

Basic reporting

The article completely satisfies basic reporting requests. I have not further comments to add apart from the minor comments highlighted in the review (please see pdf).

Experimental design

The article completely satisfies experimental design requests. I have not further comments to add apart from the minor comments highlighted in the review (please see pdf).

Validity of the findings

The article attempts to disentangle habitat vs climatic drivers influencing bird occurrence/abundance across a wide elevation gradient. This is a major and very relevant topic for climate change-related studies. The authors have correctly highlighted the novelty factor of this work throughout the manuscript.

Additional comments

Congratulation to the authors for submitting this very interesting manuscript. Most of my comments are minor (see attached pdf) and I have few major comments to make:

- Given the wise elevation gradient used I was expecting to see habitat & temperature relationships to be non-linear (i.e. quadratic) but authors have opted for linear structural equation model. I wonder if this choice might explain some of the unexpected results emerged in the study (relationship between temperature and some mountain specialist birds). Can the authors explain their rationale for choosing only linear structural equation model.

- Precipitation were not modelled in this study and authors have provided a reasonable argument why this choice is unlikely to be a major issue for the final results. However, I argue that the use of the world “climate” in the title and throughout the manuscript (i.e. “we modelled climate”) is not correct and should be changed with “temperature”.

- It will be very informative to plot mean season temperature across elevation (whilst accounting for aspect) to make the reader understand how much variation occurs across the study site .

Thanks a lot for the opportunity in reviewing this important work

Annotated reviews are not available for download in order to protect the identity of reviewers who chose to remain anonymous.

---

## Round 0.2 · accepted · Accept

As you can see, both reviewers are happy with your revision. Please consider the very minor changes suggested, though several seem unnecessary changes to the grammar.

·

Basic reporting

The basic report is good. Previous queries have been adequately dealt with. Just a few comments on the text (line numbers refer to the tracked changes Word Document)

1. L9 - "in 2018" rather than "In year 2018".
L134 - it's not clear what is meant by "are indeed hardly to occur". You mean there is hardly any evidence that topographic features have effects on alpine birds?
L168 - it should be "..not aimed at quantitatively disentangling...".
L209 - "intrepreted with caution" would be better than "cautiously considered".
L211 - I agree that chiffchaff should still be included, despite the relatively high spatial autocorrelation. The caveat provided is sufficient.

Experimental design

All is now clear - the additions to the text have clarified the authors' approach.

Validity of the findings

No further comment - all issues addressed adequately.

Additional comments

Nothing further to add.

Reviewer 3 ·

Basic reporting

NA

Experimental design

NA

Validity of the findings

NA

Additional comments

I thank the authors for the substantial changes made to the manuscript. Congratulations for your work!